# Systemic Metabolic Alterations Correlate with Islet-Level Prostaglandin E_2_ Production and Signaling Mechanisms That Predict β-Cell Dysfunction in a Mouse Model of Type 2 Diabetes

**DOI:** 10.3390/metabo11010058

**Published:** 2021-01-16

**Authors:** Michael D. Schaid, Yanlong Zhu, Nicole E. Richardson, Chinmai Patibandla, Irene M. Ong, Rachel J. Fenske, Joshua C. Neuman, Erin Guthery, Austin Reuter, Harpreet K. Sandhu, Miles H. Fuller, Elizabeth D. Cox, Dawn B. Davis, Brian T. Layden, Allan R. Brasier, Dudley W. Lamming, Ying Ge, Michelle E. Kimple

**Affiliations:** 1Department of Medicine, University of Wisconsin-Madison, Madison, WI 53705, USA; michael.schaid@northwestern.edu (M.D.S.); nicole.cummings@wisc.edu (N.E.R.); cpatiban@medicine.wisc.edu (C.P.); eringuthery@gmail.com (E.G.); austin.reuter@vumc.org (A.R.); hbrar@wisc.edu (H.K.S.); dbd@medicine.wisc.edu (D.B.D.); abrasier@wisc.edu (A.R.B.); dlamming@medicine.wisc.edu (D.W.L.); 2Interdepartmental Graduate Program in Nutritional Sciences, University of Wisconsin-Madison, Madison, WI 53706, USA; rjfenske@wisc.edu (R.J.F.); jnum@novonordisk.com (J.C.N.); 3Research Service, William S. Middleton Memorial Veterans Hospital, Madison, WI 53705, USA; 4Department of Cell and Regenerative Biology, University of Wisconsin-Madison, Madison, WI 53705, USA; yzhu353@wisc.edu (Y.Z.); ying.ge@wisc.edu (Y.G.); 5Human Proteomics Program, School of Medicine and Public Health, University of Wisconsin-Madison, Madison, WI 53705, USA; 6Department of Obstetrics and Gynecology, University of Wisconsin-Madison, Madison, WI 53715, USA; ong@cs.wisc.edu; 7Department of Biostatistics and Medical Informatics, University of Wisconsin-Madison, Madison, WI 53705, USA; 8Division of Endocrinology, Diabetes, and Metabolism, University of Illinois at Chicago, Chicago, IL 60612, USA; miles@bcg.com (M.H.F.); blayde1@uic.edu (B.T.L.); 9Jesse Brown Veterans Affairs Medical Center, Chicago, IL 60612, USA; 10Department of Pediatrics, University of Wisconsin-Madison, Madison, WI 53792, USA; ecox@wisc.edu; 11Institute for Clinical and Translational Research, University of Wisconsin-Madison, Madison, WI 53705, USA; 12Department of Chemistry, University of Wisconsin-Madison, Madison, WI 53706, USA

**Keywords:** obesity, type 2 diabetes, insulin resistance, inflammation, gut microbiome, untargeted plasma metabolomics, polyunsaturated fatty acids, prostaglandins, insulin secretion, beta-cell function

## Abstract

The transition from β-cell compensation to β-cell failure is not well understood. Previous works by our group and others have demonstrated a role for Prostaglandin EP3 receptor (EP3), encoded by the *Ptger3* gene, in the loss of functional β-cell mass in Type 2 diabetes (T2D). The primary endogenous EP3 ligand is the arachidonic acid metabolite prostaglandin E_2_ (PGE_2_). Expression of the pancreatic islet EP3 and PGE_2_ synthetic enzymes and/or PGE_2_ excretion itself have all been shown to be upregulated in primary mouse and human islets isolated from animals or human organ donors with established T2D compared to nondiabetic controls. In this study, we took advantage of a rare and fleeting phenotype in which a subset of Black and Tan BRachyury (BTBR) mice homozygous for the *Leptin^ob/ob^* mutation—a strong genetic model of T2D—were entirely protected from fasting hyperglycemia even with equal obesity and insulin resistance as their hyperglycemic littermates. Utilizing this model, we found numerous alterations in full-body metabolic parameters in T2D-protected mice (e.g., gut microbiome composition, circulating pancreatic and incretin hormones, and markers of systemic inflammation) that correlate with improvements in EP3-mediated β-cell dysfunction.

## 1. Introduction

Type 2 diabetes (T2D) is characterized by a systemic loss of blood glucose homeostasis that is primarily linked to obesity, which is often associated with insulin resistance (IR) and systemic inflammation. IR and inflammation induce stress on the pancreatic β-cell to meet the demand for enhanced insulin secretion to maintain glucose homeostasis [1]. An inability of β-cells to compensate by increasing mass, functionality, or both results in β-cell failure and hyperglycemia [2]. 

The transition from β-cell compensation to β-cell failure is not well understood. Our group and others have demonstrated a role for Prostaglandin EP3 receptor (EP3) (gene symbol: *Ptger3*), a G protein-coupled receptor (GPCR) for the arachidonic acid metabolite PGE_2_, as a significant contributor to β-cell dysfunction and loss of functional β-cell mass in T2D [3,4]. Most of our prior understanding of the islet PGE_2_/EP3 signaling pathway and its role in T2D pathophysiology utilized pancreatic islets isolated from human organ donors with T2D and mouse models with very elevated blood glucose levels compared to their lean, nondiabetic counterparts. One of the most reproducible T2D models utilizes the *Leptin^ob/ob^* mutation in the Black and Tan BRachyury (BTBR) mouse strain (BTBR^ob^) [3,4]. In these mice, islet *Ptger3* expression and PGE_2_ synthesis are dramatically upregulated, consequently suppressing insulin secretion [3,4]. Treating BTBR^ob^ islets with a specific EP3 antagonist or feeding BTBR^ob^ mice a diet low in arachidonic acid improves their glucose-stimulated insulin secretion (GSIS) response [3,4]. However, PGE_2_ has also been linked with beneficial effects on the islet phenotype. For example, PGE_2_ promotes M2 macrophage polarization, preventing pro-inflammatory cytokine production and promoting β-cell survival [5]. Therefore, what physiological changes might transition islet EP3 signaling from protective to detrimental remain unknown. 

It is well-known that BTBR^ob^ mice of both sexes rapidly and reproducibly become hyperglycemic because of underlying defects in both beta-cell function and skeletal muscle insulin sensitivity [6]. By 10 weeks of age, male BTBR^ob^ mice have end-stage diabetes, with a mean blood glucose of approximately 600 mg/dL [6]. Ten-week-old female mice have mean blood glucose levels of approximately 450 mg/dL and, by 14 weeks of age, have also progressed to end-stage diabetes [6]. The beta-cell-centric nature of BTBROb diabetes progression combined with the rapid and reproducible nature of the phenotype makes this line ideal for our studies. In the process of breeding BTBR^ob^ mice for downstream analyses, we discovered, for an approximately 6 month period, that no 10-week-old BTBR^ob^ mice in our investigator-accessible facility were hyperglycemic. This phenotype was partially related to changes in circulating metabolites after the brand of standard rodent chow was switched (explored in a different work [7]) but was not fully related to diet composition, as ultimately, the phenotype disappeared. Our previous publication found that the biggest differences in circulating metabolites in BTBR^Ob^ mice was phenotype and not diet composition and that these alterations correlated directly with beta-cell function [7]. Therefore, in this work, mice were grouped by phenotype, independent of diet. Using a discovery-based approach, we found that several metabolism-related phenotypes, including gut microbiome composition, levels of circulating pancreatic and incretin hormones, and inflammation-associated adipokines, were dramatically normalized in normoglycemic BTBR^ob^ mice compared to their T2D littermates. High-throughput untargeted metabolomics identified changes in circulating fatty acid conjugates associated with alterations in islet plasma membrane fatty acid composition downstream of PGE_2_ synthesis in plasma from T2D vs. normoglycemic BTBR^ob^ mice. Combined with a significant upregulation of islet *Ptger3* expression and the impact of an EP3-selective agonist on GSIS, our results link full-body metabolic derangements specifically with the EP3-mediated β-cell dysfunction of T2D.

## 2. Results

### 2.1. Initial Observation That Metabolic Phenotype Can Be Indpendent of Genotype in the BTBR^ob^ Mouse Model of T2D

In our work, we defined a mouse as hyperglycemic when its random-fed blood glucose level was ≥300 mg/dL, as measured using a veterinary glucometer and rat/mouse test strips: a level that should be well-exceeded by both male and female BTBR^Ob^ mice by 10 weeks of age. Unless humanely euthanized, BTBR^Ob^ mice succumb to diabetic ketoacidosis well before 20 weeks of age. We discovered a time-limited phenotype in which the random-fed blood glucose levels of our experimental BTBR^Ob^ mice did not consistently exceed 300 mg/dL until more than 20 weeks of age, with some mice living until 30 weeks of age before end-stage hyperglycemia was apparent (Figure 1). As our experimental mice were transferred from a breeding core facility to an investigator-accessible facility and other facilities across campus did not show the same phenotype (J.C.N. and M.E.K., personal observations), we hypothesized that an environmental factor must have been moderating the underlying genetic susceptibility. As a test of this factor being related to diet, BTBR^Ob^ mice were maintained on the standard-chow diet fed in the breeding core (Teklad 2020X) instead of being switched to the standard-chow diet of our investigator-accessible facility (Purina 5001). These diets have similar energy densities and macronutrient contents but differ in some micronutrients and ingredient sources (Appendix A). Severe hyperglycemia rapidly developed, with 3 of 6 BTBR^Ob^ mice were euthanized due to endpoint criteria being reached before 10 weeks of age (Figure 1B).

### 2.2. An Altered Gut Microbiota Composition Is Associated with T2D Resistance vs. Susceptiblity BTBR^Ob^ Mice

Diet can influence gut microbiome composition, and recent discoveries have consistently demonstrated the role of the gut microbiome in T2D pathology [8]. In order to determine whether an intrinsic difference in the gut microbiome composition was related to T2D phenotype penetrance, we collected fecal pellets from three 11–17-week-old normoglycemic BTBR^Ob^ mice (NGOB) and three 4-week-old BTBR^Ob^ mice transferred directly from the breeding core facility (Pre-T2D). None of the mice were severely T2D: all of the NGOB mice and one Pre-T2D mouse had blood glucose levels <300 mg/dL, while the other two pre-T2D mice had blood glucose levels <400 mg/dL (Figure 2A). 16s rRNA sequencing was performed (a complete list of the relative abundance of microbiota by phyla, class, order, family and genus between NGOB and Pre-T2D mouse fecal pellets can be found in Appendix A). Briefly, Bacteroidetes and Firmicutes composed the majority of gut microbiota, as expected, with no significant differences between the groups (Figure 2B,C). While not statistically significant, the mean total nondominant phyla abundance was reduced in Pre-T2D vs. NGOB fecal samples (Figure 2D), an effect that appeared most related to Proteobacteria abundance (Figure 2E,F). Within Proteobacteria, the Gammaproteobacteria genus comprised half in the NGOB fecal samples and was absent in Pre-T2D samples. Finally, a three-way principal component analysis (PCA) analysis of the full dataset showed clustering by groups that, while not statistically significant, was certainly supportive of a complete study (Appendix A).

### 2.3. The Gut Microbiota Composition of Normoglycemic BTBROb Mice Is Significantly Different Compared to Both T2D BTBR^Ob^ and WT Controls, At Least Partially Independent of Diet

In order to elucidate factors involved in T2D protection in the BTBR^Ob^ line, new cohorts of mice were fed the Purina or Teklad diet from 4–10 weeks of age, at which time metabolic phenotyping and terminal tissue, blood, and cecal content collection were performed. As our pilot experiments included both male and female mice, these experiments were limited to the male sex to ensure robust hyperglycemia of Teklad-fed mice at 10 weeks of age. During the course of our full dietary intervention experiment, though, the T2D-protected phenotype of Purina-fed BTBR^Ob^ mice disappeared. This provided us with the opportunity to explore factors involved in diabetes protection vs. susceptibility, at least partially independent of diet, as the WT and T2D groups were composed of both Purina-fed and Teklad-fed mice.

The mean 4–6 h fasting blood glucose levels of WT and NGOB mice were nearly identical (approximately 180 mg/dL), whereas that of T2D mice was approximately 500 mg/dL (Figure 3A). This effect was not due to differences in the Leptin^Ob^ phenotype, as both NGOB and T2D mice were similarly morbidly obese at 10 weeks of age compared to WT controls (Figure 3B). Furthermore, NGOB mice were equally as insulin-resistant as T2D littermates, with insulin having no reducing effect on blood glucose levels, whether represented as the percent change from baseline or as the area under the curve (AUC) from zero (Figure 3C).

16s rRNA sequencing was performed on cryopreserved cecal contents of 4–9 mice per group. The complete list of changes in microbial species (relative abundance) between WT, NGOB, and T2D mice and its statistical significance can be found in Appendix A. As with the pilot experiment using fecal pellets, Bacteroidetes and Firmicutes composed the majority of gut microbiota, with no significant differences among the groups (Figure 3D–F). Interestingly, phylogenetic diversity was similar between the WT and T2D groups but was significantly enhanced in NGOB animals (Figure 3G,H). Of the nondominant phyla, the relative abundance of Actinobacteria (Figure 3I), TM7 (Figure 3J), Tenericutres (Figure 3K), and Verrucomicrobia (Figure 3L) were all similar among WT, NGOB, and T2D mice. Cyanobacteria (Figure 3M) abundance was significantly lower in T2D mice compared to NGOB, and Proteobacteria abundance was substantially elevated in NGOB mice compared to both of the other groups (Figure 3N). In addition to altered total abundance, the Proteobacteria species composition in NGOB mice was vastly different compared to that of WT and T2D mice (Figure 3O, left), with specific changes in the Desulfovibrionaceae and Enterobacteriaceae families (Figure 3O, right). 

### 2.4. The T2D Phenotype Is Associated with Altered Circulating Incretin and Adipokine Levels 

Peptide hormones secreted from a number of tissues are critical for appropriate blood glucose control, and their levels are known to be dysregulated in IR and T2D [9,10,11,12,13]. Consistent with insulin hypersecretion to compensate for peripheral IR, random-fed plasma insulin levels were significantly elevated in NGOB and T2D mice compared to WT controls, with a lower mean plasma insulin level in T2D mice suggestive of emerging β-cell failure (Figure 4A).

Failure to suppress α-cell glucagon secretion is another hallmark of IR, and consistent with this, NGOB and T2D plasma glucagon levels were similarly elevated compared to WT controls (Figure 4B). A high insulin-glucagon ratio (IGR) promotes glycogenolysis and gluconeogenesis while the opposite is true for low IGR [14]. Consistent with this concept, when each mouse’s insulin value was normalized to its own glucagon value, IGR was significantly elevated in the NGOB group compared to both the WT and T2D littermates (Figure 4C). Gut incretin hormones such as glucagon-like peptide 1 (GLP-1) and gastric inhibitory polypeptide (GIP) are classically defined as potentiators of GSIS, and defects in their signaling pathways are known to contribute to the inappropriate glucose control of T2D [15]. Consistent with this concept, both GLP-1 and GIP were elevated in NGOB plasma compared to WT and even further elevated in the T2D state (Figure 4D,E). Finally, alterations in adipokine secretion due to adipose tissue meta-inflammation is becoming increasingly understood as contributing to the pathophysiology of IR and T2D [16]. The adipokines resistin and plasminogen activator inhibitor-1 (PAI-1) were both significantly elevated in T2D plasma compared to WT (Figure 4F,G). PAI-1 was mildly elevated in NGOB plasma, albeit not with statistical significance (Figure 4G). 

### 2.5. An Integrated Ultrahigh-Resolution FIE-FTCIR MS Metabolomics Approach Accurately Clusters Mouse Plasma Samples by Disease State 

We performed untargeted plasma metabolomics using a workflow integrating flow injection electrospray (FIE) with ultrahigh-resolution Fourier Transform Ion Cyclotron Resonance (FTICR) mass spectrometry (MS), a platform recently established and validated for use with cryopreserved plasma samples [7] (see Appendix A for a summary of the workflow). Between 1200 and 2200 distinct metabolic features were detected in plasma samples from each group, with over 75% of those features being annotatable by chemical structure using MetaboScape or chemical name using the METLIN Metabolite and Chemical Entity Database (Figure 5A and Appendix A). Of the 1750 distinct metabolic features, 653 were shared among the three groups (Figure 5B). A principal component analysis (PCA) performed using all detected metabolic features in both positive and negative ion modes was used to assess variability in the data (Figure 5C). Unsupervised hierarchical clustering effectively grouped plasma samples by phenotype (Figure 5D, top), with a weaker effect of diet (Figure 5D, bottom). Finally, using METLIN-annotated data, a robust increase in intensity for a hexose corresponding with a mass of 180.06351 was found in T2D plasma compared to NGOB and WT (Figure 5E): a peak we confirmed by MS/MS in a previous work to be primarily glucose, with a possible minor contribution by fructose [7]. 

### 2.6. Diet Alone Does Not Explain the Phenotype of Male BTBR^ob^ Mice at 10 Weeks of Age

Pairwise comparisons of the metabolic features of WT vs. NGOB plasma and NGOB vs. T2D plasma were performed to further validate the phenotypic predictive value of our analysis. Volcano plots were generated to elucidate significantly differentially expressed features (Figure 6A,B, left), and the top 25 most highly differentially expressed of these were again used in unsupervised hierarchical clustering analyses (full lists of the differentially expressed features between WT and NGOB plasma vs. NGOB vs. T2D plasma are shown in Appendix A, respectively). In these pairwise comparisons, an even more robust clustering of plasma samples by phenotypic group was found, with complete segregation of WT from NGOB samples and of NGOB from T2D samples (Figure 6A,B, right, top). Furthermore, while all obese mice that remained normoglycemic had been fed the Purina diet, diet alone was not predictive of phenotype, as nearly equal numbers of T2D mice had been fed the Purina vs. Teklad diet, and even in pairwise comparisons, T2D mouse plasma samples did not segregate exclusively by diet group (Figure 6B, right, bottom). 

### 2.7. FIE-FTCIR MS Metabolomics Reveals That Elevations in Circulating Eicosanoid Precursors Correlate Directly with Agonist-Dependent EP3 Signaling in T2D β-Cell Dysfunction

In our previously published work [7], the fatty acids and fatty acid conjugates family was the most highly significantly expressed for both polar and nonpolar metabolites in plasma from obese mice [7]. Eicosanoids are highly bioactive fatty acid metabolites, and a subset of essential polyunsaturated fatty acids (PUFAs) provides the substrates for their synthesis [17,18]. Using the METLIN-annotated metabolic features shown in Appendix A, the entire linoleic acid elongation and desaturation pathway was downregulated in NGOB vs. WT plasma: in most cases, with statistical significance (Figure 7A, left, gray vs. black bars). Furthermore, in all cases, the abundance of these omega-6 PUFAs was significantly elevated compared to both the other groups (Figure 7A, left, red bars). Linolenic acid serves as the initial backbone for the omega-3 PUFA subfamily, and while linolenic acid cannot be distinguished from the omega-6 PUFA gamma linolenic acid (GLA) in our analysis, its downstream products, eicosapentaenoic acid (EPA) and clupanodonic acid/docosapentaenoic acid, were also significantly upregulated in T2D plasma vs. WT (Figure 7A, right, red vs. black bars). In these cases, though, their levels in NGOB plasma were not lower than in WT, and in fact, EPA abundance was significantly elevated (Figure 7A, right).

The EP3 agonist prostaglandin E_2_ (PGE_2_), a metabolite of arachidonic acid (AA) incorporated into plasma membrane phospholipids, and PGE_2_ excretion have been found to be upregulated in islets from T2D mice and humans compared to nondiabetic controls [3,19]. EPA competes with AA for the same phospholipid position and downstream metabolic enzymes. While PGE_2_ is rapidly degraded in the blood, a specific prostaglandin E metabolite (PGEM) ELISA confirmed that elevated conjugated fatty acid levels correlated directly with increased circulation of PGE, of which PGE_2_ is by far the most abundant (Figure 7B). Finally, the baseline insulin content of islets from T2D mice was significantly reduced compared to both of the other groups (Figure 7C), consistent with β-cell failure downstream of a failed β-cell compensation response [4,6]. 

In the β-cell, the pro-inflammatory cytokine interleukin-1β (IL-1β) is known to stimulate the expression of both PGE_2_ synthetic enzymes and EP3 itself [20,21,22,23]. While IL-1β mRNA expression was nearly identical in islets from NGOB mice compared to WT, it was significantly upregulated in islets from T2D mice (Figure 7D, left). Similarly, the expression of mRNAs encoding cyclooxygenase 1 and 2 (Ptgs1 and Ptgs2, respectively), catalyzing the rate limiting step in PGE_2_ synthesis, was also significantly upregulated in islets from T2D mice compared to WT (with the latter also being higher in NGOB) (Figure 7D, middle). Finally, EP3 (Ptger3) mRNA expression was only upregulated in T2D compared to both WT and NGOB, with a fold-change of approximately 16-fold when calculated via the 2^ΔΔCt^ method (Figure 7D, right).

To confirm altered PGE_2_/EP3 signaling as a direct contributor to β-cell dysfunction in this T2D mouse model, we performed ex vivo glucose-stimulated insulin secretion (GSIS) assays with and without the addition of an EP3-selective agonist, sulprostone. Normalizing the amount of insulin secreted to total insulin content revealed enhanced insulin secretion at low glucose (1.7 mM) in NGOB islets compared to WT, consistent with the hypersecretory phenotype required to maintain fasting euglycemia, with no effect of sulprostone (Figure 7E, gray vs. black bars). In contrast, islets from T2D mice have a mild GSIS defect in response to stimulatory (16.7 mM) glucose that is even further exaggerated with sulprostone treatment (Figure 7E, red bars).

## 3. Discussion

To date, the BTBR^ob^ mouse has been understood as a model of severe T2D secondary to β-cell failure, with a consistent, full disease penetrance by 16 weeks of age in both sexes and earlier in males [4,6]. In this study, we encountered a unique cohort of male BTBR^ob^ mice for which environmental factors, including but not limited to diet, completely prevented hyperglycemia without further intervention. We have exploited these findings to study physiological changes as they pertain to islet function in obese, insulin-resistant, normoglycemic animals compared to their equally obese and insulin-resistant T2D littermates. 

The gut microbiome is of great interest in the T2D field and is well-known to be related to diet and other environmental conditions. Previous reports on mice and humans have documented alterations in the gut microbiome of T2D individuals compared to healthy controls [8,24]. While changes in the ratio of the dominant phyla, *Firmicutes* and *Bacteroidetes*, have been previously linked with T2D, we found no differences in *Firmicutes* or *Bacteroidetes* composition or relative abundance among groups [24,25]. Instead, the most defining characteristic of the NGOB gut microbiome fingerprint compared to both of the other groups was a change in nondominant phyla composition and abundance driven primarily by *Proteobacteria*. The elevations in *Proteobacteria* in NGOB mouse cecal matter are perplexing, as many reports indicate a higher disease incidence when *Proteobacteria* is elevated [26]. On the other hand, a richer overall microbial diversity has been associated with positive outcomes regarding glucose homeostasis [8,27]. 

A distinct endocrine profile was observed in the NGOB mouse compared to WT, and this profile was further altered in T2D. Elevated insulin and glucagon are hallmarks of β-cell stress and insulin resistance, and NGOB mice showed clear β-cell compensation, with a robust increase in IGR that was lost in the T2D cohort. The gut-islet relationship, of great interest in the T2D field, is thought to be modulated in part by the microbial composition of the GI tract. The incretin hormones, GLP-1 and GIP, are primarily secreted from gut enteroendocrine cells and act on the β-cell to promote insulin secretion. GLP-1 and GIP were both elevated in NGOB mice compared to WT. Surprisingly, they were even further elevated in plasma from T2D mice, even though loss of the incretin response is a known pathophysiological defect in T2D [27]. However, gut microbiota alterations have been shown to influence incretin sensitivity in obesity, insulin resistance, and T2D [28,29]; therefore, elevated GLP-1 and GIP levels may be reflective of a compensatory response that ultimately fails to promote β-cell function. 

Changes in gut microbiota are also associated with alterations in circulating factors associated with T2D pathology [28,30,31]. This relationship with microbiota dysbiosis is further supported by elevated levels of resistin and PAI-1, adipokines associated with obesity and T2D [30,32,33]. Resistin and PAI-1 production and secretion are stimulated during innate immune response and, therefore, can reflect alterations in gut endothelial cell integrity that impact the systemic metabolic profile of the organism [30,33,34,35,36]. In obesity, bacterial byproducts, such as lipo-polysaccharides (LPS), enter circulation as the endothelial wall of the intestine is degraded: a common consequence of obesity-associated intestinal inflammation [37,38]. LPS induces the release of resistin and PAI-1 from adipocytes, and LPS has also been shown to impair β-cell function via downregulation of the mature β-cell gene expression profile [39,40]. Therefore, it is possible that resistin and PAI-1 may serve as markers for β-cell dysfunction of T2D, independent of any direct influence on the β-cell themselves. More work would be necessary to tease apart this relationship.

The effects of elevated plasma AA and its precursor linoleic acid have been previously implicated in diabetes pathology and correlate with elevated HBA1c levels and hyperglycemia in human subjects [41]. In this work, we confirmed that elevated levels in circulating AA (or its isomers) correlate directly with elevated PGE metabolite levels. However, it has long been known that PGE_2_ is synthesized and excreted from pancreatic β-cells themselves and that this synthesis is a factor of both substrate availability and the expression of key synthetic enzymes, many of which are upregulated by the pro-inflammatory cytokine IL-1β [3,4,20,21,22]. In this work, we confirmed that this direct IL-1β-associated effect is of biological consequence, as the PGE_2_ analog, sulprostone, only influences the GSIS response of islets from T2D BTBR^Ob^ mice, which already exhibits a prominent secretion defect compared to their nondiabetic controls. However, even though essential PUFAs must be obtained from the diet, diet alone cannot explain the altered ratios of omega-6 and omega-3 PUFAs downstream of linoleic acid and linolenic acid (including the most important to our work, AA and EPA) as equal numbers of T2D mice had also been fed an identical diet. Still, when considered in the context of our previous work demonstrating that BTBR^Ob^ islets cultured in EPA-enriched media or nonobese diabetic mice fed an EPA-enriched diet significantly improved ex vivo or in vivo β-cell function, respectively [3], our findings certainly support the continued study of PUFA-based dietary interventions for T2D prevention or therapy.

In summary, we found that a host of systemic metabolic alterations are associated with the T2D phenotype of a strong genetic model of the disease and that, at least for the AA-to-PGE_2_ pathway, have directly implicated a specific class of metabolites in β-cell dysfunction of the disease. These findings have strong implications in the management of T2D, as many foods in the Western diet are enriched with AA. Therefore, pharmacological strategies to reduce gut/adipose inflammation in combination with a diet focused on limiting circulating AA may help to facilitate better β-cell function, either alone or in concert with T2D medications, promoting more effective blood glucose control even in the face of chronic obesity and insulin resistance. 

## 4. Materials and Methods 

### 4.1. Animal Care and Husbandry

BTBR mice heterozygous for the *Leptin^ob^* mutation were purchased from The Jackson Laboratory and bred in house at the UW-Madison Breeding Core Facility to generate homozygous OB mice or wild-type (WT) controls. Experimental mice were singly housed in temperature- and humidity-controlled environments and maintained on a 12:12 h day/light cycle with free access to acidified water (InnoVive, San Diego, CA, USA) and one of two standard mouse chows of nearly identical macronutrient composition and energy density. A direct comparison of the nutrient content and ingredients of the Teklad global soy protein-free extruded 2920X diet (Envigo, Indianapolis, IN, USA) or the Rodent Laboratory Chow 5001 diet (Purina, Neenah, WI, USA) has been previously published [7]. Both wild-type control mice (*n* = 9; 3-Purina and 6-teklad) and OB mice (*n* = 17; 11-Purina and 6-teklad) were given either Purina or Teklad diets. At 10 weeks of age, OB mice that developed hyperglycemia were grouped as T2D/HGOB (*n* = 12). A subset of OB mice was able to maintain normoglycemia, which was grouped as NGOB (*n* = 5). All protocols were approved by the Institutional Animal Care and Use Committees of the University of Wisconsin-Madison and by the William S. Middleton Memorial Veterans Hospital, which are both accredited by the Association for Assessment and Accreditation of Laboratory Animal Care (Project ID: G005181-R01-A03). All animals were treated in accordance with the standards set forth by the National Institutes of Health Office of Animal Care and Use. 

### 4.2. Blood Glucose Measurements and Insulin Tolerance Tests

Insulin tolerance tests (ITTs) were performed essentially as previously described [42]. Briefly, mice were fasted for 4–6 h and 0.75 U/kg short-acting recombinant human insulin (Humulin^®^ R; Eli Lilly, Indianapolis, IN, USA) was injected intraperitoneally. Blood glucose readings were taken by tail nick with an AlphaTRACK glucometer (Zoetis, Parsippany-Troy Hills, NJ, USA) at baseline and the indicated times after insulin administration. Percent blood glucose change from baseline was determined by normalizing the blood glucose reading at each timepoint to that at baseline for each mouse independently. The blood glucose percent change from baseline was averaged within genotypes at each time point, giving the means ± SEM.

### 4.3. Ex Vivo Islet Glucose Stimulated Insulin Secretion Assays

Islets were isolated from experimental mice at 10 weeks of age utilizing a collagenase digestion protocol as previously described [43]. Glucose-stimulated insulin secretion (GSIS) assays were performed as previously described [44] after the indicated treatments. Briefly, islets were washed and preincubated for 45 min in a 0.5% Bovine Serum Albumin (BSA) Krebs Ringer Bicarbonate Buffer (KRBB) containing 1.7 mM glucose. Islets were then incubated for an additional 45 min in either low glucose (1.7mM) or stimulatory glucose (16.7 mM) ± the EP3-selective agonist sulprostone (10 nM). Secretion media was collected, and islets were lysed in equal volume to determine insulin content. Insulin was measured via ELISA as previously described [44].

### 4.4. Quantitative PCR for Gene Expression Analyses

RNA isolation, cDNA synthesis, and quantitative PCR using SYBR Green reagent (Bio-Rad) to determine relative mRNA abundance among groups were all performed as previously described [45]. Data were normalized to that of β-actin to calculate ∆CT values. Primer sequences are available upon request.

### 4.5. Terminal Blood Collection and Plasma Hormone/Metabolite Assays

Terminal blood collection for plasma samples was performed by retroorbital puncture under anesthesia. Briefly, mice were anesthetized using 2,2,2-tribromoethanol (Sigma-Aldrich, St. Louis, MO, USA; #T48402). Blood was collected retro-orbitally using a heparin-coated glass capillary tube and mixed with 5 µM EDTA, 10 nM DDP-4 inhibitor, and 20 nM aprotinin. Plasma was isolated via centrifugation and stored at −80 °C until needed. Plasma hormones were measured utilizing the Bio-Plex Pro^TM^ Diabetes Assay (Bio-Rad Laboratories, Hercules, CA, USA) following the manufacturer’s protocol. The plasma prostaglandin E metabolite was measured utilizing a PGEM ELISA (Cayman Chemical Company, Ann Arbor, MI, USA; no. 514531) following the manufacturer’s protocol as previously described [42]. 

### 4.6. Gut Microbial DNA Preparation, Sequencing, and Analysis

Microbial 16s sequencing of cecal matter was performed as previously described [46]. Briefly, 20–50 mg of the cecal matter was collected from WT, NGOB, and T2D mice, and genomic DNA was extracted and cleaned by using the Macherey-Nagel PCR Clean-up kit according to manufacturer’s protocol (ThermoFisher Scientific, Waltham, MA, USA). Purified genomic DNA was submitted to the University of Wisconsin-Madison Biotechnology Center. DNA concentration was verified fluorometrically, and samples were prepared and amplified according to Illumina’s 16s Metagenomic Sequencing Library Preparation Protocol with few modifications as described before [46]. Following PCR, reactions were cleaned using 0.7× volume of AxyPrep Mag PCR clean-up beads (Corning, Corning, NY, USA). Quality and quantity of the finished libraries were assessed using an Agilent DNA 1000 kit (Agilent Technologies, Santa Clara, CA, USA) and Qubit^®^ dsDNA HS Assay Kit (ThermoFisher Scientific), respectively. In an equimolar fashion, libraries were pooled and appropriately diluted prior to sequencing. After sequencing, images were analyzed using the standard Illumina Pipeline, version 1.8.2. OTU assignments (Illumina, Inc., San Diego, CA, USA). Diversity plots were created using the QIIME (Ver. 1.9.1) analysis pipeline [46,47].

For the pilot experiment using fecal pellets from NGOB and pre-T2D mice, cryopreserved fecal pellets were shipped to Argonne National Labs where sequencing and data analysis were performed according to their standard protocols. Sequencing from the pilot experiment only went to the order level and not the species level.

### 4.7. FIE-FTCIR MS for Unbiased Plasma Metabolomics

Plasma samples were collected as described in Section 4.5. Samples were prepared using a methanol extraction procedure, and unbiased FIE-FTICR MS experiments were performed as previously described [7]. Statistical analysis was performed using MetaboScape 4.0 (Bruker, Billerica, MA, USA) and the online software MetaboAnalyst [48]. Putative metabolites were further annotated by METLIN with a 2-ppm mass error cutoff. Details about data analysis were described previously [7].

### 4.8. Statistical Analyses

Data from all experiments excluding that previously described for microbiome and metabolomics analyses were analyzed using GraphPad Prism v.9 (GraphPad Software Inc., San Diego, CA, USA). Data were analyzed by *t*-test, one-way ANOVA, or two-way ANOVA as described in the figure legends. *p* < 0.05 was considered statistically significant. 

### 4.9. Data Availability

All data contained within this manuscript are available upon reasonable request of the corresponding author. The mass spectrometry proteomics data have been deposited to the ProteomeXchange Consortium via the PRIDE [49] partner repository with the dataset identifier PXD022624.

## Figures and Tables

**Figure 1 metabolites-11-00058-f001:**
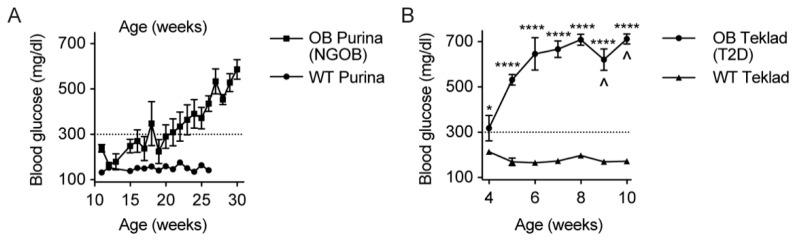
Diet partially explains the relative protection of a cohort of BTBR^Ob^ mice from early and severe Type 2 diabetes (T2D): (**A**) random-fed blood glucose readings of *n* = 6–8 Purina diet-fed wild-type (WT) and BTBR^Ob^ mice. Data represent mean ± SEM, although, as not all BG levels for all mice were recorded every week, a statistical analysis was not performed. (**B**) Weekly random-fed blood glucose readings of Teklad-fed WT and BTBR^Ob^ mice: *n* = 6–8 mice per group. Data represent mean ± SEM and were compared by two-way ANOVA with Holm-Sidak test post hoc. * *p* < 0.05 and *****p* < 0.0001. ^ indicates that 1–2 BTBR^Ob^ mice were euthanized after human endpoints were reached.

**Figure 2 metabolites-11-00058-f002:**
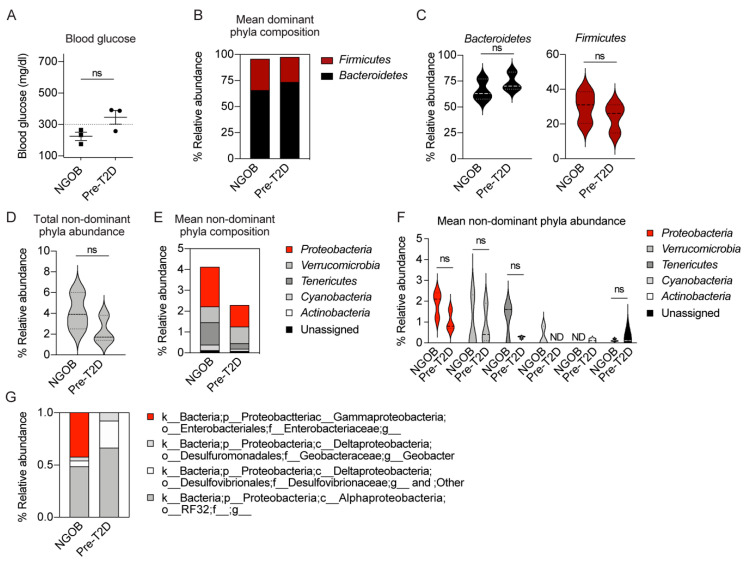
An altered gut microbiota composition is associated with T2D resistance vs. susceptibility BTBR^Ob^ mice: (**A**) random-fed blood glucose levels of two female and one male mouse per group, where data represent mean ± SEM and ns represents not significant; (**B**) bar plot of dominant phyla; (**C**) violin plots of individual dominant phyla; (**D**) violin plots of total nondominant phyla; (**E**) bar plot of nondominant phyla; (**F**) violin plots of individual nondominant phyla; and (**G**) bar plot of individual proteobacteria orders as a fraction of the total proteobacteria abundance. Data represent mean ± SD and population density. In (**C**,**D**,**F**), data were compared by *t*-test. ns = not significant.

**Figure 3 metabolites-11-00058-f003:**
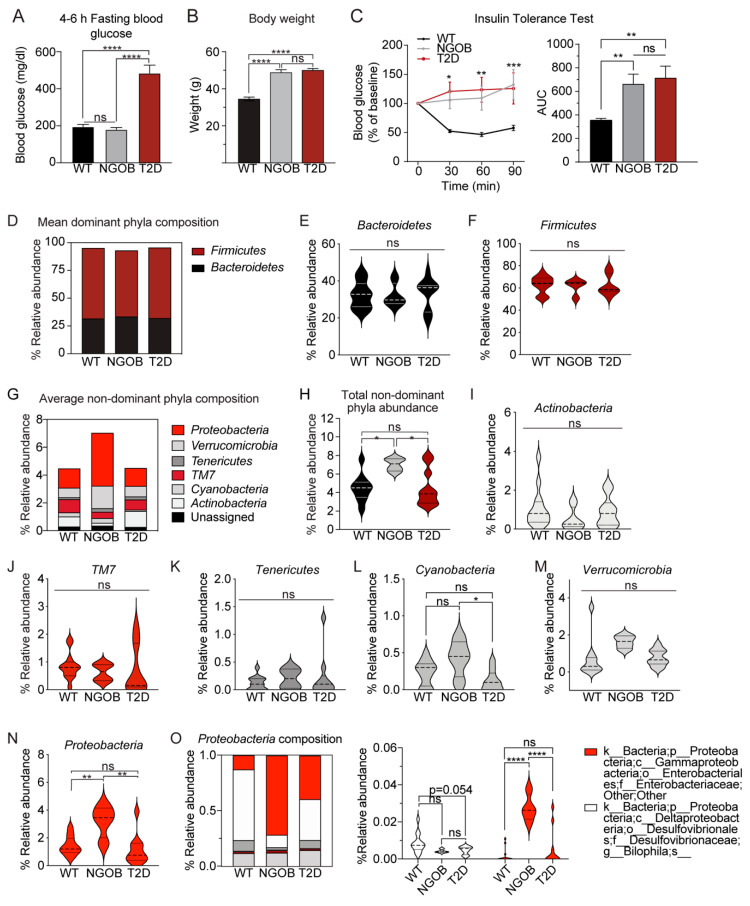
Gut microbiota diversity is associated with diabetes protection: (**A**) 4–6 h fasting blood glucose recordings of wild-type (WT) normoglycemic obese (NGOB) and diabetic (T2D) mice *n* = 6–15; (**B**) WT, NGOB, and T2D bodyweight at 10 weeks of age, *n* = 7–14; (**C**) insulin tolerance test of WT, NGOB, and T2D mice, with data plotted as a percent of baseline (t = 0 min) and quantified by the area under the curve (AUC) from zero and with data representing mean ± SEM, ns = not significant. (**D**) bar plot of dominant phyla; (**E**,**F**) violin plots of individual phyla; (**G**) bar plot of nondominant phyla; (**H**) average relative abundance of nondominant phyla; (**I**–**N**) violin plots of individual phyla; and (**O**) species composition of proteobacteria (left) and individual relative abundance of each proteobacteria species (right), *n* = 4–10. Data represent mean ± SD and population density. * *p* < 0.05, ** *p* < 0.01, *** *p* < 0.001, and **** *p* < 0.0001. ns = not significant.

**Figure 4 metabolites-11-00058-f004:**
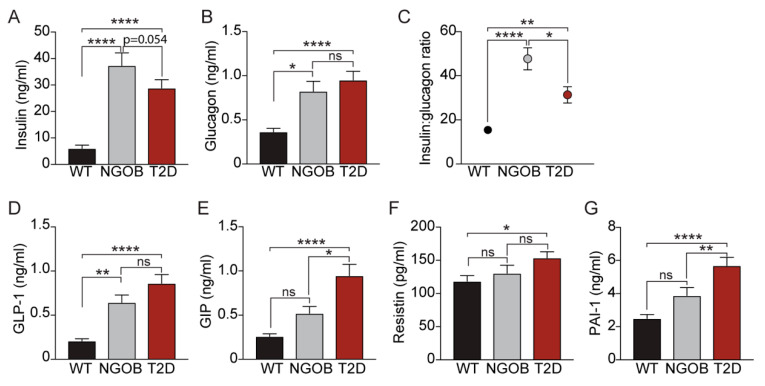
Loss of the insulin-glucagon ratio is associated with elevated incretin and adipokine levels in T2D: random-fed plasma concentrations of (**A**) insulin and (**B**) glucagon; (**C**) insulin-glucagon ratio; and random-fed plasma concentrations of (**D**) glucagon-like peptide 1 (GLP-1), (**E**) gastric inhibitory polypeptide (GIP), (**F**) resistin, and (**G**) gastric inhibitory polypeptide (PAI-1) in WT, NGOB, and T2D mice, *n* = 9–17. Data represents mean ± SEM. * *p* < 0.05, ** *p* < 0.01, and **** *p* < 0.0001. ns = not significant.

**Figure 5 metabolites-11-00058-f005:**
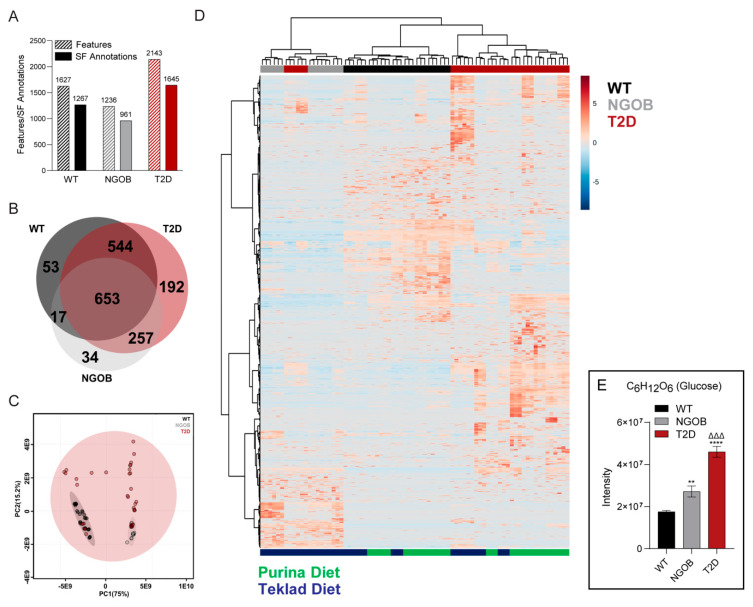
Application of flow injection electrospray (FIE)-Fourier Transform Ion Cyclotron Resonance (FTICR) mass spectrometry (MS) to plasma samples from WT, NGOB, and T2D mice: (**A**) the numbers of features (light shading) and chemical formula annotations (solid shading) from MetaboScape 4.0 (Δm < 5 ppm) of the WT, NGOB, and T2D groups; (**B**) Venn diagram of chemical formula annotations of WT, NGOB, and T2D plasma samples; (**C**) Principal component analysis (PCA) of the three groups based on FIE-FTICR MS data, where the 95% confidence limit is indicated as the shaded area; (**D**) heat map of significantly expressed metabolic features annotated by a chemical formula generated by unsupervised hierarchical clustering analysis; and (**E**) intensity of the hexose peak, experimentally confirmed to be primarily glucose, in WT, NGOB, and T2D plasma. Data represent mean ± SEM and were compared by one-way ANOVA with Holm-Sidak test post hoc. ** *p* < 0.01 and **** *p* < 0.0001 compared to WT. ΔΔΔ *p* < 0.001 compared to NGOB.

**Figure 6 metabolites-11-00058-f006:**
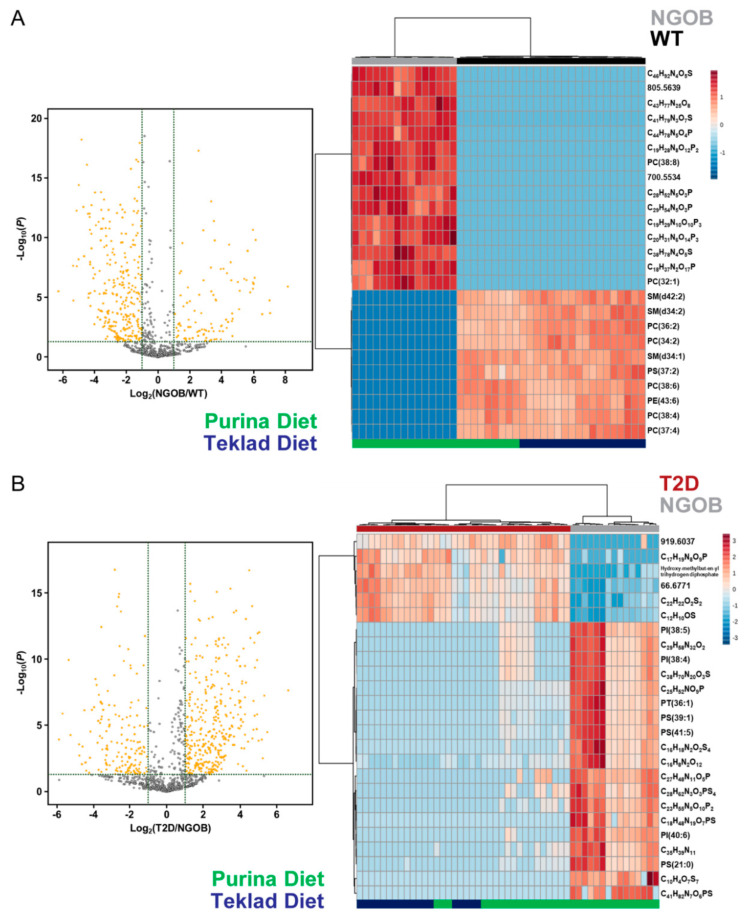
Volcano plots used to determine significantly differentially expressed annotated metabolic features (left) and heat maps created by unsupervised hierarchical clustering using the top 25 most highly differentially expressed of these (right) in pairwise comparisons of (**A**) NGOB vs. WT plasma samples and (**B**) T2D vs. NGOB plasma samples.

**Figure 7 metabolites-11-00058-f007:**
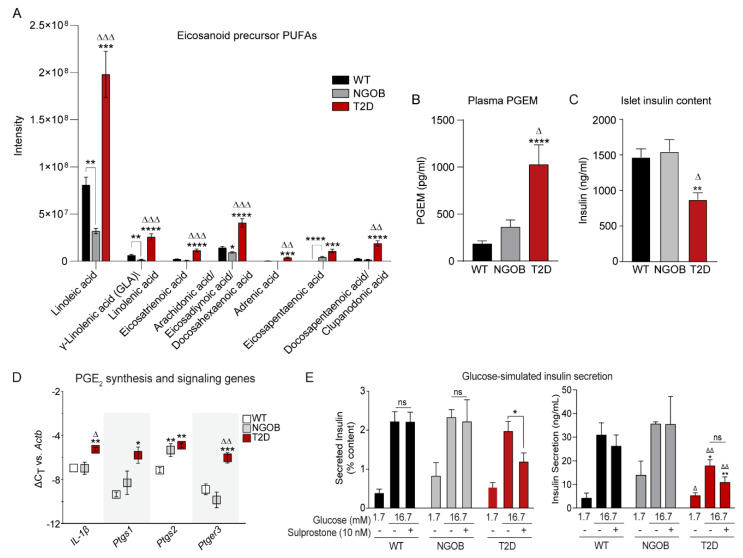
Changes in PGE_2_ production and signaling and its role in insulin secretion from WT, NGOB, and T2D BTBR mice: (**A**) comparison of significantly expressed eicosanoid precursors from WT, NGOB, and T2D plasma samples, where data were analyze by multiple *t*-test with Holm-Sidak test post hoc; (**B**) plasma PGE metabolite concentrations as measured by ELISA (*n* = 4–12); (**C**) total islet insulin content from WT, NGOB, and T2D islets, *n* = 3–12; (**D**) transcript expression of PGE_2_ synthetic and signaling enzymes from WT, NGOB, and T2D islets, where data are relative to β-actin, *n* = 4–6; and (**E**) glucose-stimulated insulin secretion from islets stimulated with 1.7 mM glucose or 16.7 mM glucose ±10 nM sulprostone. Secreted insulin was normalized to total insulin content (left). Data are plotted as total insulin secreted (right), *n* = 3–12. Data represent mean ± SEM and were analyzed by two-way ANOVA with Holm-Sidak test post hoc. * *p* < 0.05, ** *p* < 0.01, *** *p* < 0.001, and **** *p* < 0.0001 compared to WT. Δ *p* < 0.05, ΔΔ *p* < 0.001, and ΔΔΔ *p* < 0.001. ns = not significant.

## Data Availability

All data are contained within the article or Appendix A.

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
