# Peer review of "Systemic Metabolic Alterations Correlate with Islet-Level Prostaglandin E2 Production and Signaling Mechanisms That Predict β-Cell Dysfunction in a Mouse Model of Type 2 Diabetes"

_metabolites, 2021, doi:10.3390/metabo11010058_

Round 1

Reviewer 1 Report

The data presented in the present manuscript describes an interesting sub-population of the well-defined Type 2 diabetic BTBR-ob/ob mouse strain that does not develop diabetes however display all the other associated defects such as obesity, insulin resistance.  The study aimed to identify the factors that played a role to this phenotypic difference. Their analysis identified several metabolic differences between the group such as changes in the microbiome, circulating factors (glucagon, insulin, GLP-1, GIP, cytokines) and relevant to their expertise changes in EP-3 mediated beta-cell function. 

The experiments are well designed, and the data are well analysed and interpreted. The changes in the above-mentioned parameters are convincing and interesting however it is not clear whether these changes are the cause or consequence of the type 2 diabetes protection observed in the NGOB group. The most important question for me is what is the cause of this change in this subset of mice? Answering this question may be outside the scope the current manuscript but it would be good if the authors could speculate.

Other comments:

  1. It is not clear to me from the manuscript what the different diets used were and how they differ. It would be good to describe that at the beginning of section 2.5 to make it clear.
  2. Is figure 3C insulin: glucose or insulin: glucagon?
  3. What does insulin secretion in Figure 6E look if the data was not expressed relative to content? Since content in the T2D was significantly lower than the other groups would you see similar results if you express data relative to DNA content of islets?

Author Response

The data presented in the present manuscript describes an interesting sub-population of the well-defined Type 2 diabetic BTBR-ob/ob mouse strain that does not develop diabetes however display all the other
associated defects such as obesity, insulin resistance. The study aimed to identify the factors that played a role to this phenotypic difference. Their analysis identified several metabolic differences between the group such as changes in the microbiome, circulating factors (glucagon, insulin, GLP-1, GIP, cytokines) and relevant to their expertise changes in EP-3 mediated beta-cell function.

The experiments are well designed, and the data are well analyzed and interpreted. The changes in the above-mentioned parameters are convincing and interesting however it is not clear whether these changes are the cause or consequence of the type 2 diabetes protection observed in the NGOB group. The most important question for me is what is the cause of this change in this subset of mice? Answering this question may be outside the scope the current manuscript but it would be good if the authors could speculate.

We thank the reviewer for the opportunity to speculate further on the cause of the change in this subset of mice. We have added data from pilot experiments showing that a change in standard rodent chow was certainly a player in the protection phenotype, and that diet causes an intrinsic change in the gut microbiome composition prior to the development of frank T2D. These new data are presented in Figures 1, 2, and Supplemental Tables 1 and 2. Yet, diet composition is not the sole explanation for the altered gut microbiome composition, as the phenotype was lost over time. We propose another factor regulating systemic inflammation was also involved. Yet, as we cannot perform more experiments, this remains only a hypothesis. This hypothesis is supported, in part, by the changes in circulating adipokine levels (Figure 3) and pancreatic islet gene expression and function (Figure 7).

Other comments:

  1. It is not clear to me from the manuscript what the different diets used were and how they differ. It would be good to describe that at the beginning of section 2.5 to make it clear. 

With the addition of the pilot experiments, we have described the rationale for using the different diets in both the Introduction (Section 1) and the Results (Sections 2.1 and 2.2)

  1. Is figure 3C insulin: glucose or insulin: glucagon?

Figure 3C (now Figure 4C) is insulin:glucagon, which were both part of the same Bio Plex Mouse Diabetes Assay as the other hormones shown in this figure. All of these values are random-fed levels, which, for severely hyperphagic BTBROb mice are more representative of fed levels than fasting. The glucose levels shown in Figure 3A were taken after a 4-6 h fast, prior to insulin tolerance tests; therefore, we cannot combine these to perform an insulin:glucose ratio.

  1. What does insulin secretion in Figure 6E look if the data was not expressed relative to content? Since content in the T2D was significantly lower than the other groups would you see similar results
    if you express data relative to DNA content of islets?

In Figure 6E (now Figure 7E) we have presented the data as both insulin secreted as a percent of content (left) and raw insulin secretion over the 45 min of the assay (right). We cannot normalize to DNA content in our assay, but based on previous experience the significantly reduced content in T2D BTBR-Ob mice due primarily to insulin hypersecretion due to uncontrolled hyperglycemia. T2D BTBR-Ob mice do ultimately lose functional beta-cell mass, but due to the fact we are isolating islets to perform ex vivo assays, only the healthiest islets are selected, which are size-matched as much as possible.

Reviewer 2 Report

The manuscript by Schaid et. al. attempts to describe the relevance of prostaglandin E2 production in β cell dysfunction in T2D. As a model, they employed BTBR mice with alteration in the glycemic state. They correlate several phenotypes with PGE2 effect on β cell function.

 I am missing any information regarding the newly established male BTBR animals being normoglycemic obese mice (NGOB). How were these animals evolved? On which diet and breeding conditions? Why only males are involved in the study?

The manuscript is written in good English, however sometimes in compact sentences, often long and hard to follow.

The main point of the manuscript to uncover the involvement of PGE2/EP3 signaling in the transition from β cell compensation to β cell failure is not evident to me. T2D animals show clearly intact insulin secretion upon glucose stimulation in the absence of sulprostone (Fig.6E) and other phenotypes showing the failure of β cells are missing.

I have several comments to the manuscript:

  1. How do the authors explain normoglycemia in NGOB mice? They show hyperinsulinemia, no affected content of insulin in β cells and GSIS is also working fine. IR doesn´t cause further hyperglycemia compared to T2D.
  2. In many places in the discussion they rely on gut microbiota alterations, however the alterations they identified in NGOB animals cover only minor population of phyla composition with controversial effect on T2D as they admit. How do they think this is linked to improved PGE2/EP3 signaling?
  3. 3 shows plasma insulin levels, however, is it fasting insulin? What are the dietary conditions at the time of taking samples for analysis? The same for GLP-1/GIP.
  4. What does the insulin/glucagon ratio mean in terms of diabetic phenotype?
  5. Showing no variations in adipokines in NGOB means there is no chronic inflammation. Do you have any other evidence for it? Plasma concentrations of proinflammatory cytokines etc.?
  6. Concerning untargeted plasma metabolomic, I don´t understand the involvement of two diets? Can the authors explain it briefly to the text?
  7. 4E show glucose intensity in plasma, where NGOB show increased levels, however, were chosen to be normoglycemic?
  8. Chronic IL1β signalization in β cell is an important player in the deterioration of β cell function as they contain a high amount of its receptor IL1R. However, this is mainly the effect of extracellularly employed IL1β from activated macrophages. The authors show only mRNA of intracellularly expressed IL1β, which further needs inflammasome assembly to get mature to be active and ready for secretion.
  9. The authors normalize insulin secretion in Fig.6E on intracellular insulin content. However, the content doesn´t need to correspond to secretion. Can authors normalize it on DNA content?

Minor comments:

  1. Could authors specify by each experiment diet conditions of animals?

Author Response

The manuscript by Schaid et. al. attempts to describe the relevance of prostaglandin E2 production in β cell dysfunction in T2D. As a model, they employed BTBR mice with alteration in the glycemic state. They correlate several phenotypes with PGE2 effect on β cell function.

General Comments:

  1. I am missing any information regarding the newly established male BTBR animals being normoglycemic obese mice (NGOB). How were these animals evolved? On which diet and breeding conditions?

We thank the reviewer for the opportunity to clarify the rationale for our experiments and the nature of the cohorts of mice studied in this work. We have added data from pilot experiments showing that a change in the brand of standard rodent chow was certainly a player in the protection phenotype. Diet was related to an intrinsic change in the gut microbiome composition prior to the development of frank T2D. These new data are presented in Figures 1, 2, and Supplemental Tables 1 and 2. Yet, diet was not the sole explanation for altered gut microbiome composition, nor any of the other mechanisms explored in this work, as the phenotype was transient and was lost after approximately 6 months. Therefore, there is no newly-established strain.

More details will be provided below, but briefly, we propose another environmental factor was required for the full T2D protection: one that regulated systemic inflammation. Since we cannot perform more experiments, this remains only a hypothesis, although we have significant support for it. Plasma levels of circulating adipokines associated with adipose meta-inflammation (Figure 3F,G) and prostaglandin E2 metabolite, a known inflammatory mediator (Figure 7B), are both highly elevated in T2D plasma samples but not NGOB plasma samples.

  1. Why only males are involved in the study?

We have now included data from a pilot study that involved both male and female mice, and explained the rationale for including only male mice in the full study. Briefly, after showing the phenotype was penetrant in both sexes, males were chosen for further experiments to ensure consistent development of severe hyperglycemia at 10 weeks of age.

  1. The manuscript is written in good English, however sometimes in compact sentences, often long and hard to follow.

We thank the reviewer for the opportunity to clarify the writing of the manuscript.

  1. The main point of the manuscript to uncover the involvement of PGE2/EP3 signaling in the transition from β cell compensation to β cell failure is not evident to me. T2D animals show clearly intact insulin secretion upon glucose stimulation in the absence of sulprostone (Fig. 6E) and other phenotypes showing the failure of β cells are missing.

In Figure 6E (now Figure 7E) we have now presented the data as both insulin secreted as a percent of content (left) and raw insulin secretion over the 45 min of the assay (right). By including the raw secretion data, it is clear T2D BTBR-Ob islets are secreting far too little insulin than would be required to control their blood glucose levels. The BTBR strain has an underlying defect in glucose-stimulated insulin secretion (GSIS) due to a SNP in the gene for Tomosyn-2: a negative regulator of insulin granule exocytosis. Therefore, the fact we find few changes in insulin secreted as a percent of content in 16.7 mM glucose alone is not entirely unexpected, as GSIS in BTBR mice is intrinsically dysfunctional.

In considering the data presented in Figure 7E solely on its own, we agree with the reviewer that it is unclear the role PGE2/EP3 signaling plays in the transition to T2D. However, this is part of a larger body of work by our group and others’ showing the importance of PGE2 production and EP3 signaling to the beta-cell dysfunction of T2D: work that has been cited in the manuscript. Briefly, an EP3-selective antagonist, L798,106, significantly enhances GSIS in islets from T2D BTBR-Ob mice but not WT. These results alone confirm a tonic up-regulation of EP3 activation is at least partially responsible for the T2D beta-cell dysfunction of BTBR-Ob mice. In the current work, we performed the converse experiment: using an EP3-selective agonist, sulprostone, to further stimulate EP3 signaling. As shown in Figure 7E, sulprostone only has an inhibitory effect on GSIS in islets from T2D BTBR-Ob mice, but not WT or NGOB, consistent with EP3 gene expression levels.

Finally, the full effect of PGE2/EP3 signaling on T2D beta-cell function is likely not on GSIS alone, but on the ability of incretin hormones to potentiate GSIS. In previous work, we demonstrated EP3 competes for the same downstream signal transduction pathways as the cAMP-stimulatory GLP1 receptor. Sulprostone acts as a non-competitive GLP1R antagonist, reducing GLP1’s maximal potentiating effect on GSIS. Therefore, it is possible a much more dramatic secretion effect would have been observed in our ex vivo islet assays if we were to have included a stable GLP-1 agonist such as exendin-4. Due to the transient nature of the phenotype, though, further experiments cannot be performed. Even so, considering LeptinOb mice are in a constant post-prandial state due to hyperphagia, the hyperinsulinemia in NGOB mice (Figure 4A) is consistent with enhanced incretin-potentiated insulin secretion, particularly because GLP-1 and GIP plasma levels are actually higher in T2D mice vs. NGOB (Fig. 4D,E).

Specific Comments:

  1. How do the authors explain normoglycemia in NGOB mice? They show hyperinsulinemia, no affected content of insulin in β cells and GSIS is also working fine. IR doesn´t cause further hyperglycemia
    compared to T2D.

The BTBR line has two underlying genetic defects that influence T2D phenotype - skeletal muscle insulin resistance and an insulin secretion defect. Combined with a failure to up-regulate beta-cell replication genes, hyperglycemia accelerates and beta-cell failure rapidly occurs. As described above, we believe part of the protection is directly related to insulin secretion itself, whether glucose-stimulated or incretin potentiated. Plasma incretin levels are also elevated in NGOB vs. WT, though, suggesting increased incretin production may also play a role in the maintenance of euglycemia. A potential link among all of these factors may be the systemic inflammatory state, as described above. Due to the transient nature of the phenotype, we cannot perform any further experiments, but all of the evidence we do present is supportive of this concept.

  1. In many places in the discussion they rely on gut microbiota alterations, however the alterations they identified in NGOB animals cover only minor population of phyla composition with controversial effect on T2D as they admit. How do they think this is linked to improved PGE2/EP3 signaling?

This critique has been partially addressed throughout our responses to the Reviewer’s previous critiques. To expand here, we do not believe gut microbiota alterations fully explain the T2D protection phenotype, but they certainly play a part. The inclusion of a new pilot analysis and further analysis of the full 16s rRNA dataset by unsupervised heterarchical clustering and PCA analyses support a strong link between diet and microbiome composition. Yet, as the phenotype was transient, dietary-elicited changes in gut microbiome composition cannot be the sole explanation for T2D protection.

Based on unpublished data from a BioPlex Mouse Cytokine Multi-Plex Assay that used plasma from cohorts of WT BTBR mice fed the same diets during the same time period, we believe an environmentally-associated systemic inflammation phenotype was critical in BTBR-Ob mice progressing to T2D. All of the plasma from mice described in the current study has been expended; therefore, we cannot confirm this holds true in NGOB vs. T2D. Yet, there is a lot of circumstantial support for this hypothesis. The plasma adipokines, PLA-1 and Resistin, as well as Prostaglandin E2 metabolite, are all highly correlated with systemic inflammation. All three of these are significantly elevated in T2D mice vs. WT, with little-to-no elevation in NGOB plasma (Figures 4F, 4G, and 7B). Furthermore, in previously-published works from our lab, we identify the importance of pro-inflammatory cytokines (particularly IL-1β) in the beta-cell dysfunction of islets from T2D BTBR-Ob mice and its relationship to PGE2/EP3 signaling. In this work, we confirm elevated IL-1b and PGE2 synthetic gene expression in islets from T2D mice as compared to both WT and NGOB (Figure 7D), correlating directly with the GSIS response in Figure 7D.

  1. Figure 3 shows plasma insulin levels, however, is it fasting insulin? What are the dietary conditions at the time of taking samples for analysis? The same for GLP-1/GIP.

All of the hormones in Figure 3 (now Figure 4) are from plasma from random-fed animals, which, for severely hyperphagic BTBR-Ob mice, is more representative of fed levels than fasting. These hormones were measured simultaneously using a BioPlex Mouse Diabetes Multi-Plex Assay.

  1. What does the insulin/glucagon ratio mean in terms of diabetic phenotype?

An elevated insulin:glucagon ratio is directly representative of the T2D state, as when insulin is high, glucagon should be low. Furthermore, insulin must be necessarily higher in the insulin resistant state to maintain euglycemia. The insulin:glucagon ratio is 4-fold higher in NGOB vs. WT, and is dropped by half in HGOB. So, in T2D mice, insulin is failing to suppress glucagon secretion, a known defect in T2D. This finding has been better described in the text.

  1. Showing no variations in adipokines in NGOB means there is no chronic inflammation. Do you have any other evidence for it? Plasma concentrations of proinflammatory cytokines etc.?

The reviewer is correct that, with little to no variation in adipokines in NGOB mice, we expect there is no chronic inflammation in NGOB. As described above, we believe this is actually a critical factor in the protection phenotype. As described above, we do not have plasma samples left with which to measure plasma concentrations of pro-inflammatory cytokines. Yet, Prostaglandin E2 metabolite is a known inflammatory marker, and its concentration is similarly low in WT and NGOB plasma as compared to T2D (Figure 7B).

  1. Concerning untargeted plasma metabolomic, I don´t understand the involvement of two diets? Can the authors explain it briefly to the text?

The rationale behind using the two diets has now been included with the inclusion of pilot study data.

  1. 4E show glucose intensity in plasma, where NGOB show increased levels, however, were chosen to be normoglycemic?

The reviewer is correct that the glucose intensity, as quantified by untargeted plasma metabolomics, is higher in the NGOB mice as compared to WT (although not nearly to the extent as T2D) (Figure 5E). These samples were from random-fed mice, whereas the blood glucose levels shown in Figure 3A are from 4-6 h fasting mice. These results are not necessarily incongruent, as the metabolomics results likely indicated a developing glucose intolerance; a known factor in the progression to T2D.

  1. Chronic IL1β signalization in β cell is an important player in the deterioration of β cell function as they contain a high amount of its receptor IL1R. However, this is mainly the effect of extracellularly employed IL1β from activated macrophages. The authors show only mRNA of intracellularly expressed IL1β, which further needs inflammasome assembly to get mature to be active and ready for
    secretion.

IL-1β may be produced from activated macrophages, but it has been well-documented that beta-cells themselves can be stimulated to produce and secrete IL-1β, and that the majority of islet IL-1β comes from islet endocrine cells. A reference has been added to the text to support this.

  1. The authors normalize insulin secretion in Fig.6E on intracellular insulin content. However, the content doesn´t need to correspond to secretion. Can authors normalize it on DNA content?

As described above, our results cannot be normalized to DNA content, but we have included the raw secretion values, in addition to the percent secreted, in Figure 7E.

Minor comments:

  1. Could authors specify by each experiment diet conditions of animals?

A table comparing the diet composition of the Purina and Teklad diets has now been included (Supplemental Table 1). For all of the heat map generated by unsupervised hierarchical clustering, diet has been included in the legend. Yet, as described in the introduction and results, a previously-published study explored metabolomic differences in this cohort of NGOB and T2D mice by diet. In this study, we aimed to discover factors independent of diet that were involved in the phenotype.

Reviewer 3 Report

I read with great interest this study by Dr. Schaid and colleagues that uses a model of Black and Tan BRachyury (BTBR) mice homozygous for the Leptinob/ob mutation to highlighted both metabolic alterations and variations in the microbiome composition associated with the type 2 diabetes phenotype, with the arachidonic acid-PGE2 axis directly implicated in the β-cell dysfunction of the disease.

This work adds new knowledge that can help in pharmacological strategies to achieve more effective glycemic control in patients with type 2 diabetes.

Just a mistake on page 2, line 68: please replace “microphase” with “macrophage”

The study is well designed and written and worth publishing.

Author Response

I read with great interest this study by Dr. Schaid and colleagues that uses a model of Black and Tan BRachyury (BTBR) mice homozygous for the Leptin / mutation to highlighted both metabolic alterations and variations in the microbiome composition associated with the type 2 diabetes phenotype, with the arachidonic acid-PGE2 axis directly implicated in the β-cell dysfunction of the disease. This work adds new knowledge that can help in pharmacological strategies to achieve more effective glycemic control in patients with type 2 diabetes. Just a mistake on page 2, line 68: please replace “microphase” with
“macrophage”

We appreciate the Reviewer’s positive view of this work, and have corrected this spelling mistake.

Round 2

Reviewer 2 Report

I appreciate that the authors significantly improved the manuscript, especially, by explaining the used mouse model and better expressed their conclusions by already obtained data. However, the model itself (NGOB mice) would deserve better phenotypization in respect to other known used mouse models for diabetic research.  The delay in glucose intolerance onset and/or transient nature of phenotype, switching of diet, give rise to more questions than are solved in the manuscript. This will produce sole incomplete study hard to compare with the knowledge in the literature using standard mouse models. Authors state many times in responses that they cannot perform more experiments which would support their results and suggestions. For example, suggested role of PGE2/EP3 on incretin role in insulin secretion of NGOB animals in vivo might be shown by OGTT +/- sulprostone or more direct data showing environmentally-associated systemic inflammation are missing and according to the authors cannot be performed. Raw data of insulin secretion didn´t bring any additional information as one can suggest that T2D animals have fewer islets thus the lower secretion of insulin is expected (haven´t found any number of islets in methods).